# Integrating Egocentric and Robotic Vision for Object Identification Using Siamese Networks and Superquadric Estimations in Partial Occlusion Scenarios

**DOI:** 10.3390/biomimetics9020100

**Published:** 2024-02-08

**Authors:** Elisabeth Menendez, Santiago Martínez, Fernando Díaz-de-María, Carlos Balaguer

**Affiliations:** 1System Engineering and Automation Department, University Carlos III, Av de la Universidad, 30, 28911 Madrid, Spain; scasa@ing.uc3m.es (S.M.); balaguer@ing.uc3m.es (C.B.); 2Signal Theory and Communications Department, University Carlos III, Av de la Universidad, 30, 28911 Madrid, Spain; fdiaz@ing.uc3m.es

**Keywords:** human–robot interaction, gaze, Siamese network, image matching, superquadrics, pose estimation, primitive shapes

## Abstract

This paper introduces a novel method that enables robots to identify objects based on user gaze, tracked via eye-tracking glasses. This is achieved without prior knowledge of the objects’ categories or their locations and without external markers. The method integrates a two-part system: a category-agnostic object shape and pose estimator using superquadrics and Siamese networks. The superquadrics-based component estimates the shapes and poses of all objects, while the Siamese network matches the object targeted by the user’s gaze with the robot’s viewpoint. Both components are effectively designed to function in scenarios with partial occlusions. A key feature of the system is the user’s ability to move freely around the scenario, allowing dynamic object selection via gaze from any position. The system is capable of handling significant viewpoint differences between the user and the robot and adapts easily to new objects. In tests under partial occlusion conditions, the Siamese networks demonstrated an 85.2% accuracy in aligning the user-selected object with the robot’s viewpoint. This gaze-based Human–Robot Interaction approach demonstrates its practicality and adaptability in real-world scenarios.

## 1. Introduction

In today’s world, robots have become increasingly common in both everyday life and industry. They are no longer limited to performing simple and repetitive tasks. Now, they must understand and respond to users’ needs more effectively [1,2]. This shift in expectations has highlighted the significance of Human–Robot Interaction (HRI). Within this context, gaze interaction emerges as an intuitive and non-verbal method of communication. Gaze is naturally intuitive as it constitutes a fundamental aspect of human interaction that is universally understood across different cultures and languages [3]. This universal understanding makes gaze an effective and straightforward communication tool without the need for speech or significant learning. Gaze enables rapid communication of intentions, ideal in dynamic settings. It enhances collaborative interactions by allowing robots to better understand and anticipate human actions [4]. By integrating gaze-based situational awareness, robots can respond appropriately to user focus, thus improving the overall interaction quality. Furthermore, the non-verbal nature of gaze communication offers accessibility in various scenarios while reducing the cognitive load for users [5].

Recognizing the importance of gaze in HRI for collaborative or assistive tasks, its application in pick-and-place tasks is crucial. Accurately understanding where the user is looking is essential for an effective interaction. Some methods use the robot’s camera [6], but they are limited to scenarios where both the objects and the user’s face are visible. In other works, the user wears eye-tracking glasses [7] equipped with two cameras: one dedicated to detecting the pupils and another capturing the user’s viewpoint. By projecting the gaze tracking onto the image seen by the user, an understanding of the user’s visual intent is gained. Most work employing eye-tracking glasses in HRI focuses on gaze-based intention estimation for interactions like grasping with objects in fixed positions or known to the robot’s object detector [8,9]. While interpreting gaze and intent is undeniably crucial, the subsequent challenge is substantial: translating this information into meaningful robot actions, especially when the robot lacks prior knowledge of the objects in its environment. How does the robot discern which object the human is focusing on and intending to interact with? Few HRI studies with eye-tracking glasses have focused on identifying where the user is looking in the robot’s image. Ref. [10] uses markers for translating the gaze from the eye-tracking glasses image to the robot’s image, and [11] employs feature descriptors, facing challenges in scenarios where the robot is facing the user or varying viewing angles.

This article presents a novel HRI approach in which the robot identifies and grasps objects based solely on the user’s gaze using eye-tracking glasses. This method is especially beneficial for individuals with limited mobility, cognitive issues, or language impairments who manage their daily routines alone, whether at home or in a hospital setting. By accurately determining the user’s gaze target object and its location relative to the robot, the approach provides an intuitive and accessible means of interaction for these users. It effectively functions across diverse viewing conditions without external markers, fixed object locations, or a limited set of objects recognized by the robot. Figure 1 illustrates the scenario this work focuses on; a user equipped with eye-tracking glasses selects an object on a table using their gaze. The robot’s task is then to identify and pick up the selected object, bringing it closer to the user. In the depicted scenario, the user freely moves and spontaneously selects an object with their gaze, focusing on a red glass in this instance. The robot then must deduce that the user’s intent is directed towards this glass. However, a challenge arises in this setting; the robot is unaware of the objects present in its vicinity or their specific locations, making the task of identifying the red glass and determining its location non-trivial. This article introduces a strategy that enables a robot to identify objects based on gaze estimations, even in the absence of prior knowledge about the objects. The solution employs a category-agnostic pose and shape estimator based on superquadrics for robust object recognition, focusing on their pose and shape regardless of category. Additionally, Siamese networks are utilized to correlate the object in the user’s gaze, captured through eye-tracking glasses, with the objects in the robot’s field of view from its camera, ensuring accurate identification of the target object.

Building upon this gaze-based object identification system, future research could investigate the robot’s ability to anticipate additional user needs related to the selected object, thereby enhancing the system’s predictive capabilities for more sophisticated and context-aware assistance. Following this introduction, the background of related work is presented in Section 2. Section 3 provides an overview of the proposed solution. Section 4 details the category-agnostic object shape and pose estimation method. The use of Siamese networks for the robot’s identification of gazed objects is explained in Section 5. In Section 6, experimental findings that highlight the robustness and accuracy of the approach in real-world scenarios are presented. Finally, Section 7 draws conclusions and discusses the future work.

## 2. Related Work

### 2.1. Object Pose and Shape Estimation

Current methods predominantly use deep convolutional neural networks to estimate the pose of objects. Labbe et al. [12] utilized RGB images and mesh models to determine an object’s pose relative to the camera. Ref. [13] employed a sequence of RGBD images to track the pose and size of novel objects, bypassing the need for a CAD model. Similarly, ref. [14] inferred the 3D poses of known objects from a single RGB image. Ref. [15] inferred 3D poses of unseen objects from an RGBD image. However, these methods often require CAD models, are limited to known or category-specific objects, and can be time intensive.

In scenarios with primitive-shaped objects where there is no need to know the category of them, superquadrics offer a less time-consuming and more versatile solution. Ref. [16] presented a rapid method for determining the shape and pose of individual objects from single-viewpoint cloud data by fitting superquadrics, utilizing a multi-scale voxelization strategy. Ref. [17] developed a method for grasping unknown symmetric objects in clutter using real-time superquadric representations. By leveraging superquadric parameters, their approach quickly determines object dimensions and surface curvature, offering efficient and accurate grasping in cluttered environments. Ref. [18] utilized superquadrics to model the graspable volume of a humanoid robot’s hand and the object, enabling real-time grasping of unknown objects without collisions. In [19], they enhanced their superquadric-based object modeling and grasping method by integrating prior shape information from an object classifier. Lastly, ref. [20] presented a probabilistic method to recover superquadrics from point clouds of a single object obtained from multiple views.

### 2.2. Human–Robot Interaction Based on Gaze

In Human–Robot Interaction, understanding gaze-based intention, especially for tasks like grasping or manipulating objects, is a critical focus area. Researchers use eye-tracking devices to capture gaze data, which comprises the eye’s position as 2D coordinates on a calibrated surface or within a scene. These data, crucial in revealing gaze events such as fixations and saccades, have been extensively studied, as noted in the work of Belardinelli et al. [8]. To decode and understand these gaze data, various predictive models have been employed. These include Hidden Markov Models, which Fuchs [21] has shown to be effective in pick-and-place tasks, and advanced recurrent neural networks like LSTM, used by Gonzalez-Díaz et al. [22] and Wang et al. [23]. Their work demonstrated the use of LSTM networks in conjunction with gaze data to predict complex human actions like grasping, reaching, moving, and manipulating objects.

Gaze-based intention estimation is vital in HRI for understanding a user’s intent to interact with objects. The challenge, however, extends beyond this. It is essential for the robot not only to identify the specific object the user intends to interact with using eye-tracking glasses but also to determine its location in the robot’s reference frame. To address this, Weber et al. [10] proposed a method that guides robot interactions based on the user’s visual intentions. It merges gaze data with images from the robot’s camera, allowing the robot to recognize objects in its own reference frame. Additionally, they developed techniques for combining gaze data with automatic object location proposals. These techniques facilitate the identification of interaction objects without requiring category-specific knowledge. However, this technique’s reliance on external markers in the environment for alignment limits its versatility.

Continuing to explore alternative methods, [24] focused on a novel approach using human gaze and augmented reality (AR) in human–robot collaboration. Their method allows robots to identify and learn about unknown objects by acquiring automatically labeled training data, thereby enhancing their object detection capabilities. They utilize Virtual Reality (VR) glasses to create a shared-gaze-based multimodal interaction. This interaction allows users to see from the robot’s perspective.

Furthermore, the approach by Hanifi et al. [6] introduces an innovative system. The system employs the robot’s camera for face detection, human attention prediction, and online object detection, thereby interpreting human gaze. This method enables the robot to accurately establish joint attention with human partners, enhancing interaction in collaborative scenarios. However, it is important to highlight that this system requires both the user’s face and the objects to be within the robot’s camera frame. This limitation restricts its utility in real-world scenarios. This fact underscores the need for more flexible HRI solutions, especially in environments where the user or objects may not always be within the robot’s field of view.

In their research, Shi et al. [11] focused on projecting human gaze from eye-tracking glasses to the robot’s camera image, using invariant feature descriptors for this purpose. However, their study revealed challenges in scenarios with significant viewpoint changes. Continuing their research, Shi et al. in [9] introduced GazeEMD, a method that uses Earth Mover’s Distance (EMD) to compare hypothetical and actual gaze distributions over objects. GazeEMD operates by running object detectors in both the user’s and the robot’s viewpoints and then matches labels to identify objects within the robot’s viewpoint. While this approach improves traditional gaze detection methods and enhances accuracy and robustness, it is limited to objects already recognized by the robot.

In summary, current research in gaze-based HRI predominantly focuses on understanding user intent in manipulation tasks. However, there is a notable gap in identifying the selected object using gaze data within the robot’s reference frame. Existing methodologies often rely on external markers [10] or invariant feature descriptors [11], which is less effective in certain scenarios with drastically different viewpoints between the user and the robot. Alternative methods using object detectors [9] are limited to objects already recognized by the robot. The proposed approach introduces a novel application of Siamese networks in this context [25], allowing for the effective matching of gazed objects in the user’s perspective to the robot’s perspective without these limitations. Notably, this approach is unique in its methodology and application, differing significantly from existing methods in gaze-based HRI. As such, direct comparison with other systems is not viable due to the distinct nature of this approach.

### 2.3. Siamese Networks

Given the complexities in correlating objects across varying perspectives, recent research has gravitated towards leveraging deep convolutional neural networks (CNNs) that adeptly learn relevant features directly from images. Within this domain, Siamese networks have distinguished themselves. These networks are uniquely engineered to learn embeddings that effectively capture the similarity between images [25]. This capability renders them particularly suitable for patch matching tasks, where discerning subtle similarities is crucial.

The utility of Siamese networks extends across various applications, demonstrating their versatility and effectiveness. They have been successfully employed in image matching tasks performed on landmark datasets [26], face verification processes [27], plant leaf identification [28], and visual tracking [29], among other areas. These applications highlight the networks’ ability to handle a range of image recognition and correlation challenges, making them a robust choice for complex image processing tasks.

Considering their adaptability, precision, and robustness, Siamese networks offer an effective solution for correlating objects across different perspectives in real-world environments.

## 3. Overview of the Proposed Solution

The proposed methodology integrates a series of distinct stages to accurately determine the object targeted by a user’s gaze, all within the robot’s reference frame. This innovative approach eliminates the need for external markers or predefined object locations. Figure 2 shows the integrated workflow of the proposed solution. The Tiago++ robot [30], equipped with an Asus Xtion RGB-D sensor [31], captures the scene. This stage utilizes depth and RGB images from the sensor, as detailed in Section 4. The process focuses on identifying the shapes and poses of objects on a horizontal surface using superquadrics estimation, intentionally bypassing the need to categorize them. Each object is assigned a unique ID number, facilitating easier identification throughout the process. Bounding boxes, corresponding to the perceived shapes of individual objects, are meticulously extracted from the RGB image captured by the robot’s camera.

Simultaneously, the user, equipped with eye-tracking glasses, begins by looking away from the objects to prevent premature gaze fixation. As the process starts, the user then navigates the environment and is free to fixate on any object at their discretion. The Pupil Invisible glasses [32] feature a camera that captures the visual field at 30 Hz and an eye-tracker that records the user’s gaze at 120 Hz. The gaze intention estimation module operates in real time, synchronized with the 30 Hz video feed. It analyzes these data to determine the user’s intention to grasp a specific object, providing a decision probability. When this probability surpasses a predefined threshold, this module triggers an action, supplying a cropped image around the gaze point from the user’s viewpoint, along with the object category. While the specific workings of this module are beyond the scope of this article, its real-time output is vital for the robot’s responsive processing in Human–Robot Interaction.

In cases where the robot completes the shape and pose estimation process before the user’s intention to grasp reaches the threshold, it enters a wait state. Upon receiving the trigger, the system rapidly matches the cropped image of the gazed object with the robot’s current view, identifying the object intended for interaction. This is achieved through the Siamese network for the robot’s process for identification of the gazed object, as detailed in Section 5. The process uses a Siamese network to extract feature vectors from both the user’s cropped view and the crops obtained from the robot camera’s bounding boxes. It then compares these feature vectors, selecting the crop most similar to the user’s view as the gazed object. Once the most similar crop is obtained, the corresponding superquadric is identified. This is possible because each crop derived from the robot’s view is associated with a unique ID number, linked to its respective superquadric. Thus, identifying the crop not only pinpoints the gazed object but also provides its superquadric model, including shape and pose. Conversely, if the user’s intention to grasp an object reaches the threshold before the robot finishes the superquadric estimation, the cropped image of the gazed object is stored temporarily. Once the robot completes the superquadric estimation, it proceeds to match the stored image with the robot’s view to identify the gazed object.

The subsequent sections concentrate on two vital components: “category-agnostic object shape and pose estimation” and “Siamese network for robot’s identification of gazed object”. These sections detail the processes critical for the robot’s identification of gazed objects under partial occlusions.

## 4. Category-Agnostic Object Shape and Pose Estimation

This section describes the steps for estimating the shape and pose of the objects in the robot’s reference frame using only one depth image acquired with the robot’s camera under a few assumptions. First, the objects are situated on a horizontal surface. Second, the objects can be partially occluded by other objects but they must not be placed on top of each other. Third, the objects can be more or less modeled with primitive 3D shapes. These primitive shapes are 2D shapes on the horizontal plane extended on the vertical axis. Finally, the robot’s camera looks down at the objects from an inclined angle.

The procedure begins with the transformation of the depth image into a 3D point cloud, followed by the removal of the horizontal surface to isolate the objects. Subsequently, the point cloud is segmented into clusters, each representing a distinct object. These clusters are then reconstructed. Finally, the reconstructed clusters are fitted into superquadric models, which serve as masks in the subsequent matching process with the RGB image.

### 4.1. Object Cloud Segmentation and Reconstruction

The initial step involves using a depth image acquired with the robot’s camera, as depicted in Figure 3. The depth image is combined with the camera’s intrinsic parameters and the robot’s joint configuration. Forward kinematics convert the depth image into a 3D point cloud in the robot’s reference frame. The limitations of depth images result in certain parts of the objects being on the occluded side of the camera or partially occluded by other objects. The method addresses this issue by assuming that the objects in the dataset can be roughly modeled as primitive 3D shapes. This assumption aids in reconstructing the point cloud, including the occluded parts.

The 3D point cloud contains points from the objects and the table where they are situated. RANSAC [33] is used to identify points corresponding to the table surface. Once identified, these points are removed from the point cloud, retaining only those associated with the objects on the table. The remaining points are grouped into distinct object clusters using Euclidean Clustering based on the horizontal coordinates. Clusters smaller than a minimum threshold, likely representing noise, are filtered out. Figure 4 shows the remaining clusters that correspond to the objects on the table. A unique label ID is assigned to each cluster to distinguish between the different objects.

After obtaining the clusters corresponding to the objects on the table, the next step is to reconstruct the occluded parts. This is performed by projecting the points of each cluster onto the horizontal plane and computing their convex hull [34]. The resulting polygon contains a set of points in counter-clockwise order that represents the 2D shape of the object. The 2D shape of the object, defined as a set of points, is extruded along the *z*-axis. This is performed by adding a constant z-value from the z-value of the table plane (the height of the table) to the maximum z-value of the cluster (the highest point of the object). This results in a set of points that forms a simplified 3D representation with no top or base. To generate the base and top points, the centroid of the 2D shape is computed first as the average of all points in the shape. Next, the original 2D shape from the centroid is scaled using multiple scaling factors, lower than 1, to add more points to the base and the top. Scaling the original 2D shape with factors lower than one generates points inside the original 2D shape. The sets of points of the base and the top are made equal, and a constant z-value is assigned to each set of points to represent the base and the top of the object.

The final step combines the extruded points with the generated base and top points, resulting in a simplified 3D point cloud that captures the essential geometric features of the object. This simplified representation provides a comprehensive view of the object, accounting for any occluded parts and emphasizing its overall shape and height. This simplified point cloud enables faster and less computationally demanding superquadric fitting in the next step compared to using a denser point cloud. Figure 5 depicts the simplified 3D point clouds, represented as colored cubes, derived from the segmented point clouds shown in Figure 4.

### 4.2. Superquadric Fitting

Superquadrics provide a compact representation of simple objects using a set of parameters. They are represented by the inside–outside function (see Equation (Equation 1)), which considers an object-centered frame.
(1)$$\begin{array}{c} F\left(x,y,z\right)={\left({\left(\frac{x}{\lambda_{1}}\right)}^{\frac{2}{\lambda_{5}}}+{\left(\frac{y}{\lambda_{2}}\right)}^{\frac{2}{\lambda_{5}}}\right)}^{\frac{\lambda_{5}}{\lambda_{4}}}+{\left(\frac{z}{\lambda_{3}}\right)}^{\frac{2}{\lambda_{4}}} \end{array}$$

In this equation, the parameters $\lambda_{1}$, $\lambda_{2}$, and $\lambda_{3}$ represent the lengths of the semi-axes along the *x*-, *y*-, and *z*-axis, respectively. The shape parameters $\lambda_{4}$ and $\lambda_{5}$ modify the curvature of the surface, influencing the shape’s overall roundness. $\lambda_{4}$ affects the curvature along the *z*-axis, while $\lambda_{5}$ affects it along the *x*- and *y*-axis. The ability of these parameters to create various shapes, from spheres to prisms, gives the superquadric model the versatility to represent diverse simple geometries. The inside–outside function evaluates a given set of coordinate points in 3D space (x,y,z). This function is needed to establish a relationship of a point with respect to the superquadric surface. If F<1, the point is inside the superquadric; if F=1, it is on the surface; and if F>1, the point is outside the superquadric. The superquadric function can be represented in the robot’s reference frame by adding three variables for translation ($p_{x},p_{y},p_{z}$) and three RPY angles (θ,ϕ,γ) for orientation.

The process of representing an object with a superquadric involves determining the best-fitting parameter vector ($v=[\lambda_{1},\lambda_{2},\lambda_{3},\lambda_{4},\lambda_{5},p_{x},p_{y},p_{z},\phi ,\theta ,\gamma ]$). These parameters are computed to align the superquadric model closely with the simplified point cloud obtained in the previous step. This is achieved by solving a constrained optimization problem that minimizes the distance between the point cloud and the superquadric surface. The objective is to adjust v so that most of the points ($p_{i}=\left[x_{i},y_{i},z_{i}\right]$) in the point cloud lie on, or very close to, the superquadric surface. Equation (Equation 2) describes the minimization problem.
(2)$$\begin{array}{cc} \; & \min_{v}\sum_{i=1}^{N}{\left(\sqrt{\lambda_{1}\lambda_{2}\lambda_{3}}\left(F(p_{i},v)-1\right)\right)}^{2}\\ \mathrm{subject}\;\mathrm{to}:\\ \; & \left\{\begin{array}{c} \left|\theta \right|\leq \epsilon \\ \pi /2-\epsilon \leq \theta \leq \pi /2+\epsilon \\ -\pi /2-\epsilon \leq \theta \leq -\pi /2+\epsilon \end{array}\right.\\ \; & 0\leq \gamma <2\pi \\ \; & \left|\phi \right|\leq \epsilon \end{array}$$

In this equation, *N* represents the number of points in the point cloud, and ϵ is a small value that constrains the orientation parameters (ϕ,θ,γ). These constraints ensure a stable and accurate representation of symmetric objects, particularly when they are placed on a horizontal surface. To solve this Sequential Quadratic Programming (SQP) problem, the method utilizes the open-source NLopt library [35]. The computation is further optimized by fitting superquadrics to each object’s point cloud in parallel using threads, significantly reducing processing time. Figure 5 visually demonstrates the effectiveness of this method, where the superquadrics, depicted as colored forms, accurately fit the simplified point cloud of each object, represented by the colored cubes.

### 4.3. Mask Based on Superquadric

The surface of a superquadric in the local frame can be represented by a set of points obtained using its direct formulation [36]. Equation (Equation 3) describes a point (${\;}^{L}p_{i}={[}^{L}x_{i}{,}^{L}y_{i}{,}^{L}z_{i}]$) on the surface of the superquadric given the semi-axes lengths $\lambda_{1}$, $\lambda_{2}$, and $\lambda_{3}$, and the shape parameters $\lambda_{4}$ and $\lambda_{5}$, as well as the iteration variables η and ω.
(3)$${\;}^{L}p_{i}=\left(\begin{array}{c} \lambda_{1}\cdot cos{\left(\eta \right)}^{\lambda_{4}}\cdot cos{\left(\omega \right)}^{\lambda_{5}}\\ \lambda_{2}\cdot cos{\left(\eta \right)}^{\lambda_{4}}\cdot sin{\left(\omega \right)}^{\lambda_{5}}\\ \lambda_{3}\cdot sin{\left(\eta \right)}^{\lambda_{4}} \end{array}\right)$$
where $-\frac{\pi }{2}\leq \eta \leq \frac{\pi }{2}$, and −π≤ω≤π.

The points of the superquadric in the local frame are transformed into the camera frame. Let ${\;}_{L}^{R}T$ be the transformation matrix from the local frame to the robot reference frame and ${\;}_{R}^{C}T$ be the transformation matrix from the robot reference frame to the camera frame. The points on the superquadric surface in the camera frame (${\;}^{C}p_{i}={[}^{C}x_{i}{,}^{C}y_{i}{,}^{C}z_{i}]$) can be obtained by Equation (Equation 4).
(4)$${\;}^{C}p_{i}{=}_{R}^{C}T\cdot_{L}^{R}T\cdot^{L}p_{i}$$

Then, the method computes the closest point ${\;}^{C}p_{closest}$ on the superquadric to the camera origin. This computation involves finding the point on the superquadric surface that has the shortest Euclidean distance to the camera origin. The distance between the closest point and the camera origin is stored for each superquadric.

The points of each superquadric surface in the camera frame are transformed into image coordinates $\left(u_{i},v_{i}\right)$ using the pinhole camera model (Equation (Equation 5)).
(5)$$\left(\begin{array}{c} u_{i}\\ v_{i}\\ 1 \end{array}\right)=\left(\begin{array}{ccc} f_{x} & 0 & x_{0}\\ 0 & f_{y} & y_{0}\\ 0 & 0 & 1 \end{array}\right)\left(\begin{array}{c} {\;}^{C}x_{i}\\ {\;}^{C}y_{i}\\ {\;}^{C}z_{i} \end{array}\right)$$
where $f_{x}$ and $f_{y}$ are the focal lengths in the *x*- and *y*-axis, and $(x_{0}$, $y_{0})$ is the principal point.

After transforming the points of each superquadric surface in the camera frame to image coordinates using the pinhole camera model, the bounding box is computed by finding the minimum and maximum values of the transformed coordinates along each axis. Specifically, the bounding box for the superquadric is computed as $\left(u_{min},v_{min},u_{max},v_{max}\right)$ in pixels, where the top-left corner corresponds to the minimum values $\left(u_{min},v_{min}\right)$, and the bottom-right corner corresponds to the maximum values $\left(u_{max},v_{max}\right)$. The bounding boxes are used to know the location of each object in the robot’s image. After computing the bounding boxes, the image region of each superquadric is cropped using its respective bounding box coordinates. To manage occlusions, the concave hull of each superquadric in the camera’s view is derived. Black fills are applied to the concave hulls of occluding objects closer to the camera, while, for more distant objects, regions encompassed by the union of the concave hull intersection and the concave hull of the objects themselves are filled. This process effectively masks out both occluding and background entities within each object’s bounding box, ensuring the object of interest is distinctly highlighted. Subsequently, the image region of each object is cropped using its respective bounding box, resulting in isolated images of each object with occluding and background elements masked out. Figure 6 showcases the processed images, with areas of other objects blackened, leaving only the object of interest visible.

## 5. Siamese Network for Robot’s Identification of Gazed Object

This section describes the method for computing the gazed object in the robot’s reference frame when all objects in the scene are different. This approach uses a Siamese network to match the patch around the gazed object in the glasses image with the most similar patch around each object in the robot image. The Siamese network is trained to learn the similarity between images of gazed objects captured with the eye-tracking glasses and images from the robot’s camera. This approach provides practical advantages for real-world scenarios since it avoids the need for markers. Using two separate object detectors for the glasses and robot images proves both inefficient and redundant. In contrast, the Siamese network offers an optimized alternative. The Siamese network is notable for its ability to identify similarities based on feature representations, requiring fewer training images. Additionally, these networks can adapt to new objects with minimal retraining. In many cases, they can operate effectively without the need for further retraining.

### 5.1. Siamese Network Framework

The Siamese network framework is specifically designed to determine the similarity of inputs. Siamese neural networks can learn similarity metrics and contain multiple (usually two or more) identical sub-networks (often referred to as branches). These branches have the same configuration and share parameters and weights. By using an identical network (or branches) to process the inputs in a Siamese network, the network learns to generate similar embeddings for similar inputs or dissimilar embeddings otherwise.

Triplet Networks extend the concept of Siamese networks. Their design focuses on learning embeddings from a triplet of samples: an anchor, a positive, and a negative image. The anchor and positive images are similar, while the anchor and negative images are dissimilar. As depicted in Figure 7, the training procedure of a Triplet Network contains several steps. It starts with the dataset preparation, followed by the sampling triplets of images. These samples are then fed into the Triplet Network. The last step involves computing the loss function to update the network’s weights. During the training phase, triplet samples are fed into the Siamese network with the goal of minimizing a specific loss function. The selection of the sampling strategy from the training dataset and the choice of an appropriate loss function are critical. These selections ensure efficient training and avoid meaningless computations.

#### 5.1.1. Triplet Neural Network

The proposed Triplet Network comprises three branches, each with a convolutional neural network, a spatial pyramid pooling layer, and several fully connected layers, as depicted in Figure 8. Notably, the last convolutional layer and the first fully connected layer have a spatial pyramid pooling layer between them in each branch. This architecture enables the network’s capability to extract and encode features from images of different sizes. The convolutional neural network in each branch acts as the feature extraction module, with the feature maps being the output of these convolutional layers. These feature maps are subsequently processed by the spatial pyramid pooling layer, which divides them into a set of fixed-size grids and pools the features in each grid separately. Finally, the fully connected layers generate the feature embeddings, which are compared to compute the loss during the training phase.

The feature extraction module uses the ResNet-50 architecture as the backbone of the Triplet Network, without the global pooling layer and the fully connected layer [37], as depicted in Figure 8. ResNet-50 is a popular deep neural network architecture that starts with a convolutional layer and a maximum pooling layer. It then progresses through four main stages, each containing a specific number of residual blocks (ResBlocks); the first stage has three blocks, the second stage has four blocks, the third stage contains six blocks, and the final stage has three blocks. Within these ResBlocks, the convolutional layers have filters that increase in number through the stages, starting from 64 and doubling through each stage up to 512. The key innovation of ResNet-50 is the incorporation of skip connections or shortcuts that connect non-adjacent layers to address the vanishing gradient problem.

The spatial pyramid pooling (SPP) layer [38] enables the output of fixed-length feature vectors regardless of the input image sizes. As depicted in Figure 9, the SPP layer operates at three levels. It processes the input feature maps, which are the output of the last convolutional layer of the ResNet-50 architecture, consisting of 2048 feature maps. The SPP divides these feature maps into grids of varying sizes. These grids are determined by pooling window sizes of [1 × 1, 2 × 2, and 4 × 4]. In SPP, a pooling window refers to the region of the feature map that is pooled together to produce a single value. By pooling over these different window sizes, the SPP layer produces 21 bins (each bin being a pooled region of the feature map). This pooling results in a fixed-length size of 21 × 2048. This ensures that the network can handle inputs of varying sizes and consistently outputs feature vectors of a predetermined length, as required by the triplet loss function. The resulting feature vectors are then passed through three fully connected layers, which learn to map the features to the embedding space.

#### 5.1.2. Sampling Strategy and Loss Function

The Triplet Network, during its training phase, learns to optimize a triplet loss function [39]. The core idea of this function is to refine the network’s ability to distinguish the distance between the anchor and the positive images while maximizing the distance between the anchor and the negative images. For a given input *x*, let *f*(*x*) represent the embedding produced by a single branch of the Triplet Network. Given a batch *i* of N triplet samples $x_{a}^{i}$, $x_{p}^{i}$, $x_{n}^{i}$, where $x_{a}^{i}$ is the anchor image, $x_{p}^{i}$ is the positive image, and $x_{n}^{i}$ is the negative image, the triplet loss function is defined as:(6)$$\begin{array}{c} L_{tri}=\sum_{i=1}^{N}max\left(0,d\left(f\left(x_{a}^{i}\right),f\left(x_{p}^{i}\right)\right)-d\left(f\left(x_{a}^{i}\right),f\left(x_{n}^{i}\right)\right)+\alpha \right) \end{array}$$
where $d(f\left(x_{a}^{i}\right),f\left(x_{p}^{i}\right))$ denotes the squared Euclidean distance between the embeddings of $x_{a}^{i}$ and $x_{p}^{i}$, and $d(f\left(x_{a}^{i}\right),f\left(x_{n}^{i}\right))$ denotes the squared Euclidean distance between the embeddings of $x_{a}^{i}$ and $x_{n}^{i}$. The term α≥0 acts as a margin to ensure a clear separation between the pairs ($x_{a}^{i}$, $x_{p}^{i}$) and ($x_{a}^{i}$, $x_{n}^{i}$).

Triplets can be classified based on the relative distances between the anchor, positive, and negative samples. A triplet is considered “hard” if the distance between the anchor and the positive is greater than the distance between the anchor and the negative, meaning that the negative is closer to the anchor than the positive. On the other hand, a triplet is considered “easy” if the distance between the anchor and the positive is smaller than the distance between the anchor and the negative, meaning that the positive is closer to the anchor than the negative. In addition, triplets can also be classified as “semi-hard” if the distance between the anchor and the positive is smaller than the distance between the anchor and the negative but the difference between the two distances is smaller than a predefined margin. Intuitively, being told repeatedly that images of an object taken with really similar viewpoints are the same object does not teach the network anything. However, seeing images of the same object from really different viewpoints or similar-looking but different objects dramatically helps it to understand the concept of similarity. The selection of triplets during training is crucial for the effectiveness of the network as training only using easy triplets can lead to a rapid decrease in the triplet loss and slow down the training process. While selecting hard triplets would make the learning process more efficient, relying only on hard triplets will lead to a network that struggles to distinguish standard triplets.

An effective method for computing the triplet loss is the batch hard strategy, as proposed in [40]. This approach constructs batches through a random selection process. In each batch, *P* different classes (or objects) are selected, and then *K* images from each class are stored. These images are collected using both the eye-tracking glasses and the robot’s camera, resulting in a total of 2PK images in each batch.

For each image *a* in the batch taken with the eye-tracking glasses, the batch hard strategy involves identifying the most challenging positive and negative images from those captured with the robot’s camera. The “hardest” positive is the farthest same-class image from the anchor within the batch, and the “hardest” negative is the closest different-class image to the anchor. This approach ensures that the triplets formed are the most informative for training the network. The batch hard triplet loss is computed using Equation (Equation 7).
(7)$$\begin{array}{c} L_{BH}=\sum_{i=1}^{P}\sum_{a=1}^{K}\left[\alpha +\max_{p=1\ldots K}D\left(f{(}^{G}x_{a}^{i}),f{(}^{R}x_{p}^{i})\right)\right)\\ -\min_{\begin{array}{c} j=1\ldots P\\ n=1\ldots K\\ j\neq i \end{array}}D{\left(f{(}^{G}x_{a}^{i}),f{(}^{R}x_{n}^{j})\right]}_{+} \end{array}$$

In this equation, ${\;}^{G}x_{a}^{i}$ and ${\;}^{R}x_{p}^{i}$ represent the *a*-th image taken with the glasses of the *i*-th object in the mini-batch, respectively. The function *D* measures the distance between the embeddings of these images. The batch hard triplets selected in this way are considered “moderate” in difficulty; they are the hardest within their mini-batch. Training with such triplets is ideal for the triplet loss as it ensures that the network learns to discern subtle differences between images.

### 5.2. Application of the Siamese Network for Robot’s Identification of Gazed Object

This subsection details the application of the Siamese network in determining the most similar object in the robot’s reference frame, corresponding to the user’s gaze selection. The process begins with cropped images of the objects in the robot’s color image. These crops are computed based on superquadrics, as detailed in Section 4.3. Each crop is linked to a unique identifier corresponding to its superquadric, ensuring precise object localization.

Simultaneously, the gaze intention estimation module, previously introduced in the overview of the proposed solution (Section 3), provides a real-time crop of the object that the user is gazing at through the eye-tracking glasses. This module also supplies the object’s category, although the specific workings of this module are not the focus of this publication.

The core of this application is a single branch of the trained Triplet Network, which generates embeddings for both the crops from the robot’s image and the crop from the eye-glasses image. The process then involves calculating the squared Euclidean distance between the embedding of the gazed object (from the eye-tracking glasses image) and the embeddings of the crops in the robot’s image. The robot’s image crop with the smallest distance to the gazed object’s embedding is identified as the object the user intends to interact with.

This identification process yields not only the category of the selected object, as provided by the gaze intention estimation module, but it also retrieves its shape and location relative to the robot’s reference frame. For a visual representation of this integrated workflow, refer to Figure 2 in the overview section.

## 6. Experiments and Results

This study conducted thorough experiments to validate two key components: the category-agnostic object shape and pose estimation (Section 4) and the Siamese network for the robot’s identification of a gazed object (Section 5). These components are pivotal in accurately determining the shape and pose of all objects within the robot’s reference frame and in pinpointing the specific object the user is gazing at. This dual capability enhances the robot’s interaction potential in environments with partial occlusions. The following sections present detailed evaluations, demonstrating the practical effectiveness of each component in real-world robotic applications.

The initial set of experiments (Section 6.1) assessed the robustness of category-agnostic object shape and pose estimation under various partial occlusion scenarios. This phase is critical for demonstrating system effectiveness in diverse real-world conditions. The experiments involved multiple objects in different occlusion contexts, focusing on the accuracy of superquadric representations for object shape and pose.

Following this, the validation of the Siamese network is presented in two distinct phases. In the first phase, detailed in Section 6.2.1, the goal was to select the most effective network architecture. This was achieved by testing on a dataset containing crops from both user and robot perspectives without occlusions. This step is essential for identifying the network architecture that most accurately matches the user’s gazed object as perceived in the robot’s view. The second phase, further elaborated in the Section 6.2.2, involved evaluating the chosen network architecture in scenarios with partial occlusions. Here, the comparison was conducted between two sets of crops, both derived using the category-agnostic shape and pose estimation process: one set as standard crops and the other modified with black masks. This phase aimed to demonstrate the improvement in object matching accuracy achieved by employing black masks, thereby simplifying the process and avoiding the need for extensive training on datasets with complex inter-object occlusions.

Finally, the efficiency of executing both processes was evaluated, particularly in situations where rapid object selection by the user is critical. This aspect, covered in a dedicated subsection, is of utmost importance in HRI contexts, where quick and accurate response times are essential for practical applications.

### 6.1. Category-Agnostic Object Shape and Pose Estimation

The accurate estimation of object shape and pose, especially under partial occlusions, is pivotal for enhancing the robustness of robotic perception in real-world scenarios. In the conducted experiments, 12 distinct objects were subjected to partial occlusions across 200 different cases. In each case, four objects were strategically placed such that they partially occluded others, creating a variety of visibility conditions for the objects involved. This setup aimed to investigate the robustness and accuracy of the proposed method under partial occlusions, which is crucial for real-world applications where objects of interest are often not fully visible.

Figure 10 presents examples of different experimental scenarios, illustrating the variety of shapes and poses of the objects involved. On the left, the color images of the scenes are shown, while, on the right, the point clouds with superquadrics superimposed as colored forms are depicted. Notably, these examples demonstrate that the objects are accurately represented by the superquadrics in terms of both shape and pose, even in the presence of partial occlusions.

The method presented in Section 4, enables the estimation of the shape and pose of the objects without knowing their category. However, for the purpose of validating the accuracy of the obtained shapes in the experiments, the objects were manually assigned with categories. This categorization allowed comparison of the estimated superquadric parameters with the actual known shapes of the objects.

Table 1 summarizes the mean and standard deviation of the superquadric parameters for different object categories, computed under a variety of occlusion scenarios. These parameters, specifically, $\lambda_{1}$, $\lambda_{2}$, $\lambda_{3}$, $\lambda_{4}$, and $\lambda_{5}$, characterize the superquadric representation of each object and serve as a metric to evaluate the performance of the proposed method in estimating the shape of partially occluded objects. The occlusion scenarios were crafted by placing objects in diverse positions and orientations to simulate real-world visibility challenges. For error computation relative to the actual shape, the ground truth parameters of all objects were obtained. The error was computed as the distance error between the point cloud generated by ground truth information and the point cloud generated by the obtained superquadric. A small error not only indicates accurate shape estimation but also implies precise determination of the object’s position and orientation given that the fitting process yields the superquadric parameters alongside its position and orientation.

A noteworthy observation based on the superquadric parameters summarized in Table 1 is the low standard deviation exhibited by $\lambda_{1}$, $\lambda_{2}$, and $\lambda_{3}$ across all object categories. These parameters, which define the sizes of the superquadric axes, are crucial for accurate shape representation. The low variability in these parameters indicates that the proposed method consistently estimates the principal dimensions of the objects, even amidst various occlusion scenarios. Conversely, the shape parameters $\lambda_{4}$ and $\lambda_{5}$ exhibit higher standard deviations, indicating a more variable estimation across the tested scenarios, as seen in Table 1. Particularly, $\lambda_{5}$ demonstrates notably high variability for objects with shorter, cylindrical shapes. This variability might be attributed to the resolution of the utilized point cloud in detecting the object shape as $\lambda_{5}$ transitions from representing a more parallelepiped, rounded shape to a more cylindrical form. The resolution of the point cloud could potentially influence the accuracy with which smaller or more intricate shape details are captured, thereby affecting the consistency of the shape parameter estimations.

### 6.2. Siamese Network

In this section, a detailed exploration of the Siamese network’s evaluation is presented, expanding upon the two key phases introduced earlier. The first part of the analysis focuses on selecting the most effective network architecture, as detailed in Section 6.2.1, involving tests with a dataset comprising images from both user and robot perspectives without occlusions. The second part, outlined in Section 6.2.2, examines the network’s performance in scenarios with occlusions, with a particular focus on the impact of black masks derived from superquadric estimations. These comprehensive analyses assess the network’s performance and accuracy under varied and practical scenarios.

#### 6.2.1. Training and Evaluating Different Architectures

A custom dataset, without occlusions and consisting of common breakfast foods, was assembled to train and evaluate the Siamese network. This dataset comprises 18 distinct objects, with 10,000 images captured using the eye-tracking glasses and another 10,000 taken with the robot. This dataset was split into training (70%), validation (15%), and test (15%) sets. Figure 11 displays sample images from both eye-tracking glasses and the robot camera. Images from the eye-tracking glasses often show motion artifacts due to the user’s natural head movements while gazing at objects. In contrast, the robot’s images, taken from a static position, offer a consistent view. The Siamese network, trained on these images, is designed to manage the challenges imposed by motion artifacts and varying perspectives.

The branches of the Siamese network, as explained in Section 5, are designed to recognize image similarities and dissimilarities through a feature extraction module, a spatial pooling layer, and several fully connected layers. While the architecture described in the aforementioned section uses ResNet-50 as the feature extraction module, ResNet-18, VGG16, VGG19 [41], and MobileNet-v2 [42] were also evaluated. Notably, all five architectures were assessed without their original fully connected layers.

Each variant of the Siamese network, utilizing ResNet-50, ResNet-18, VGG16, VGG19, and MobileNet-V2 as feature extraction modules, was trained using PyTorch [43]. An Adam optimizer with a fixed learning rate of 0.0001 was employed. The batch hard triplet loss, as previously detailed in Section 5.1.2, was employed as the loss function. This involved constructing batches by selecting three classes (P) and six images per class (K) for both glasses and robot images. To manage GPU memory constraints while maintaining a substantial batch size, gradient accumulation was employed over two steps. This strategy increased the effective batch size to 72 (3 classes × 6 images/class × 2 image types × 2 accumulation steps). Early stopping, stopping training when validation loss increased despite a decreasing training loss, was used to mitigate overfitting. Table 2 details all these hyperparameters, including the specifics of the training strategy and loss function.

In contrast to the training phase, the evaluation in the test was assessed using a triplet-based approach, where each triplet consists of an anchor, a positive, and a negative image, without specifically targeting the hardest examples. This approach evaluates the model capability to discern similar images under typical conditions. The evaluation metrics in the test set, including average loss, accuracy, precision, recall, and F1 Score, are presented in Table 3. These metrics were computed employing the optimal threshold, selected within a range of 0.1 to 2, to categorize image pairs as similar or dissimilar based on the distance between their embeddings. ResNet-50 was chosen as the preferred feature extraction module, demonstrating superior performance across all metrics, and particularly excelling in achieving the highest accuracy and F1 Score, thereby indicating balanced precision and recall and establishing itself as a proficient model in identifying similar and dissimilar images. Resnet-18, VGG16, and VGG19 exhibited suboptimal accuracy and F1 Score. MobileNet-V2, although computationally efficient, compromised precision, risking higher false positive rates, despite its acceptable recall. While suitable for resource-limited applications, its precision trade-off may misclassify dissimilar images as similar.

#### 6.2.2. Evaluation and Enhancement under Partial Occlusions

In this subsection, the Siamese network, utilizing ResNet-50 as the feature extraction module in its branches, is evaluated under conditions of partial occlusions. This architecture was selected as the most effective among various networks evaluated without occlusions in the preceding experiments. Utilizing the same dataset of 200 cases, as referenced in Section 6.1, the impact of black masks (detailed in Section 4.3) on the Siamese network’s performance under partial occlusions is explored.

The evaluation was executed by constructing triplets, the anchor being the image of the object captured with the glasses, the positive being the crop of the same object taken with the robot, and the negative being a crop of a different object. The robot-derived crops were utilized in two distinct manners: direct crops and crops enhanced with superquadric-based black masks. Given that each of the 200 cases contains four objects, the evaluation comprised 800 crops from the robot’s perspective, for both direct and mask-enhanced approaches, paired with images of the objects taken with the glasses.

Table 4 shows the performance metrics of the Siamese network, contrasting scenarios with and without superquadric-based masks. The utilized threshold, 1.5970, aligns with the one established in the preceding evaluation, ensuring metric consistency across evaluations. While the accuracy and F1 Score without occlusions were 0.97 and 0.89, respectively, these metrics exhibited a decline under partial occlusions. Notably, the incorporation of superquadric-based black masks attenuated this reduction, affirming their utility in enhancing the network’s robustness and reliability under partial occlusions and thereby sustaining viable performance metrics in practical applications.

### 6.3. Execution Time

The execution time for object identification in the system, assessed on a 12th Gen Intel^®^ Core™ i7-12700H CPU with an Nvidia GeForce RTX 3070 Ti GPU, involves two key stages. Initially, the robot computes the shape and poses of objects in parallel threads, averaging 40 ms per object. During this phase, the user starts by looking away. As the test begins, they are free to fixate their gaze on any object at their discretion. Upon gaze fixation, the system uses a Siamese network to match the gaze to an object in the robot’s view, taking an additional 4 ms. Thus, the total time from the user’s gaze fixation to object identification is approximately 4 ms, plus the initial 40 ms for robot computation if not already completed. This efficient response time underscores the real-time capability of the system in Human–Robot Interaction scenarios.

## 7. Conclusions

The primary goal of this work was to develop a system that integrates egocentric and robotic vision for accurate object identification based on the user’s gaze. The methodology employs a category-agnostic estimator for determining the shape and pose of all objects on a table, using superquadrics without relying on object categories. The Siamese network is then applied to determine which of these objects, each characterized by unique shape and pose, aligns with the user’s gaze. This process is assisted by an external gaze intention estimation module, which provides additional insights into the object’s category and the user’s intention to interact.

The effectiveness of this approach has been validated across various scenarios, particularly in environments with partial occlusions. The shape and pose estimator demonstrated high precision, with an average error of only 8 mm in basic shape estimation. The Siamese network achieved a notable 97.1% accuracy in non-occluded settings and maintained an 85.2% accuracy in occluded environments, all without the need for retraining the network for occluded images.

A key strength of the approach is its adaptability to new objects, which can be integrated without necessitating retraining of the Siamese network from scratch. While occasional fine-tuning may be needed to optimize performance, this flexibility is highly beneficial for real-world scenarios. Furthermore, the methodology eliminates the need for external markers or specific object positioning in the environment. Unlike traditional object detection systems, which might require a separate detector on the robot and depend on label comparisons, the proposed approach is not limited to the object categories used in training. This adaptability makes the approach particularly suitable for a variety of real-world scenarios.

However, it also presents several limitations. The system’s accuracy may diminish in highly complex scenes with numerous objects or severe occlusions. Factors such as variability in lighting conditions and visual noise can impact performance. Challenges in differentiating objects with similar appearances and the reliance on accurate gaze tracking and consistent user behavior also pose potential issues. Additionally, the system assumes the uniqueness of each object in the scene, primarily offers basic shapes for objects, and presupposes that all objects are directly situated on the table.

Looking ahead, three key areas of future work have been identified. First, there is a need to extend the current approach to accurately estimate the poses and shapes of stacked objects. This advancement requires improvements in point cloud segmentation and superquadric optimization processes, crucial for handling real-world scenarios with stacked objects. Second, improving the identification process of the object being gazed at by the user is essential. The current system uses a single image crop from the user’s viewpoint compared with one crop per object from the robot’s perspective. To enhance this process, it is proposed to incorporate multiple image crops from the user’s perspective, taken from several frames before and after the user’s command. This multi-frame integration will allow a more detailed comparison, improving the system’s precision and accuracy. Lastly, the robot estimates the poses and shapes of objects based on one single depth image. However, to enhance the system’s capabilities, there is a need to develop a process where the robot dynamically updates its estimations of objects’ shapes and poses while navigating through an environment. As the robot moves, it can capture additional images, which can be used to refine the point cloud and update the estimations of object shapes and poses in real time.

## Figures and Tables

**Figure 1 biomimetics-09-00100-f001:**
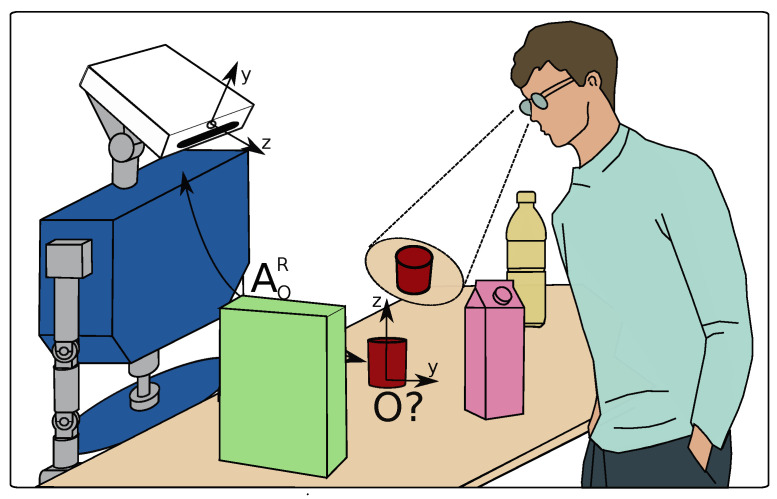
The user gazes at a red glass on a table, prompting the robot to identify and grasp it. However, the robot lacks prior knowledge of the object’s categories and positions.

**Figure 2 biomimetics-09-00100-f002:**
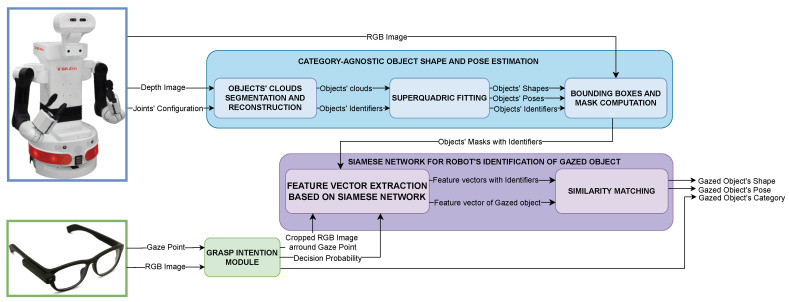
Workflow of the gaze-based object identification process.

**Figure 3 biomimetics-09-00100-f003:**
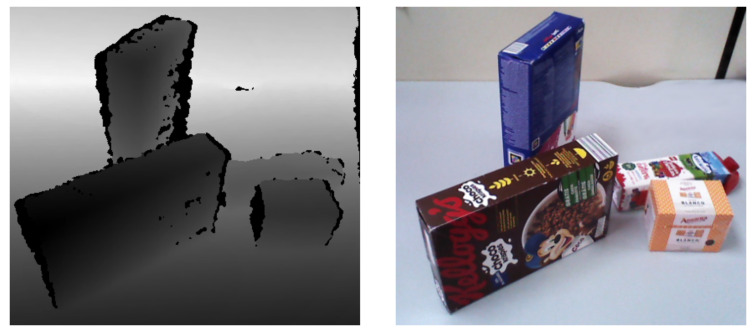
Depth and color images from the robot’s camera display partially occluded objects.

**Figure 4 biomimetics-09-00100-f004:**
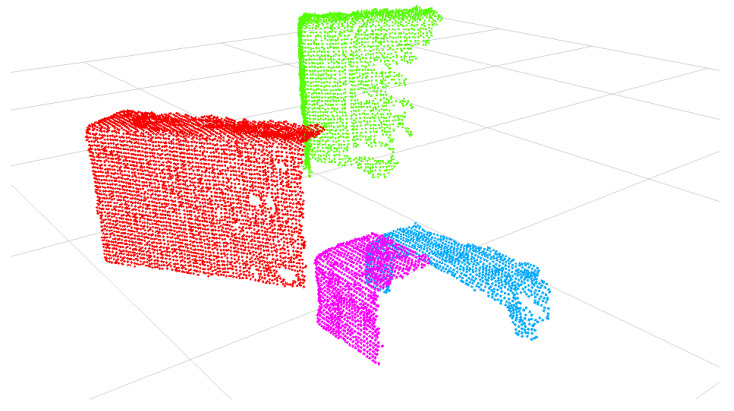
Each object cluster is assigned a different label ID.

**Figure 5 biomimetics-09-00100-f005:**
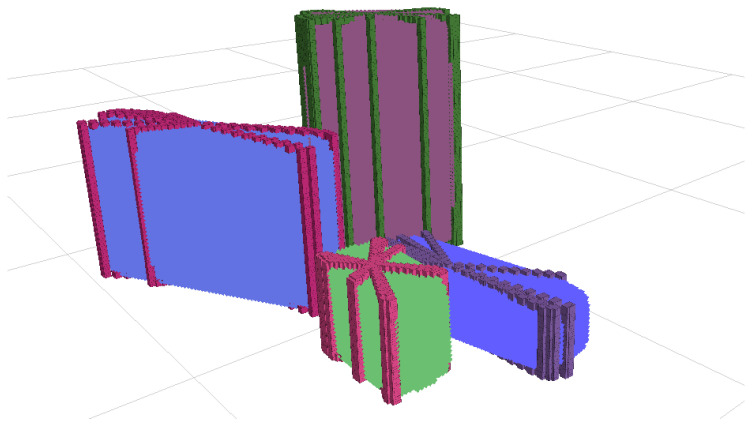
Simplified 3D point clouds represented as colored cubes and superquadric fits depicted as colored forms, demonstrating shape and pose estimation of objects.

**Figure 6 biomimetics-09-00100-f006:**
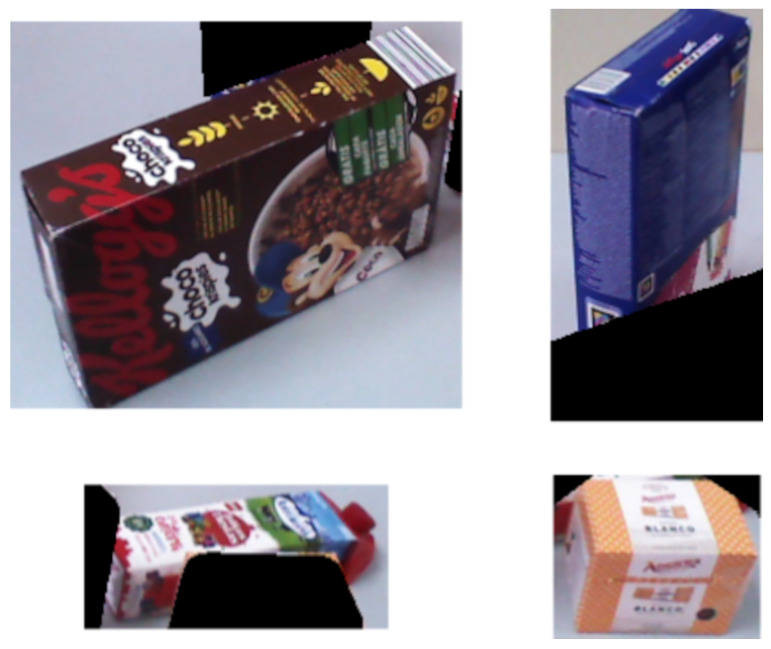
Crop of each object with other objects blacked out.

**Figure 7 biomimetics-09-00100-f007:**
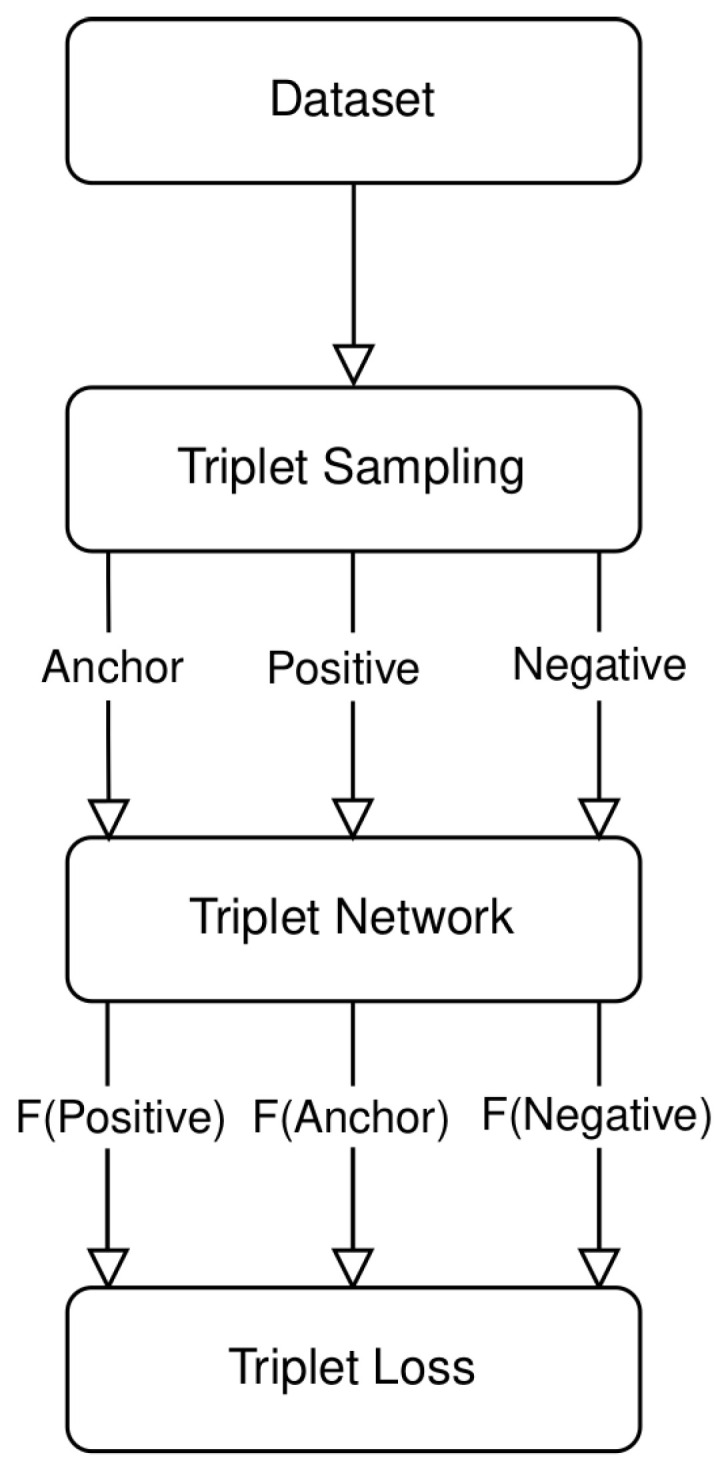
Overview of the general procedure for training the Siamese network.

**Figure 8 biomimetics-09-00100-f008:**
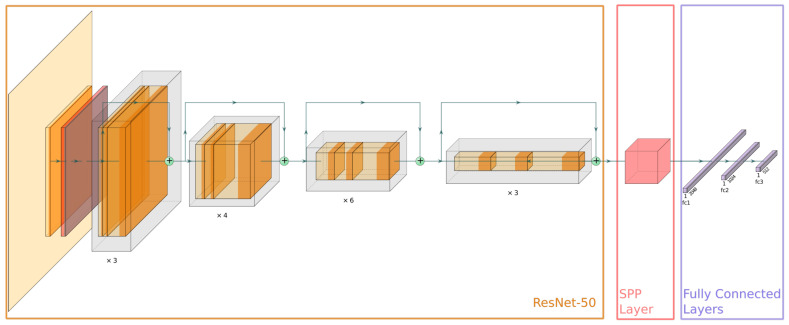
Architecture of the Siamese network branch with ResNet-50, spatial pyramid pooling layer, and fully connected layers for feature extraction.

**Figure 9 biomimetics-09-00100-f009:**
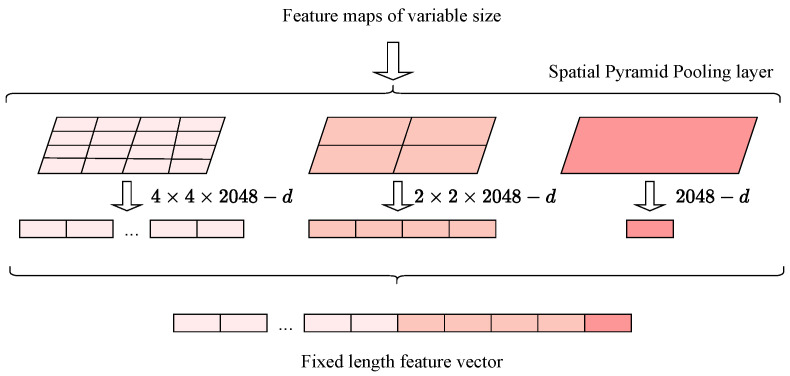
Illustration of an SPP layer with three levels. The number of feature channels at each level is represented by ‘d’, indicating depth.

**Figure 10 biomimetics-09-00100-f010:**
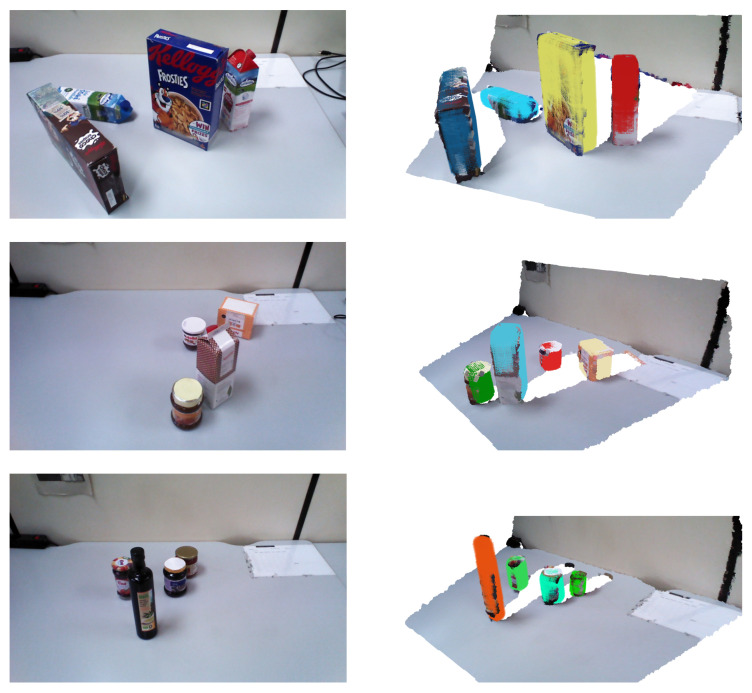
(**Left**)—color images of scenes with partially occluded objects; (**Right**)—point clouds with superimposed superquadrics as colored forms.

**Figure 11 biomimetics-09-00100-f011:**
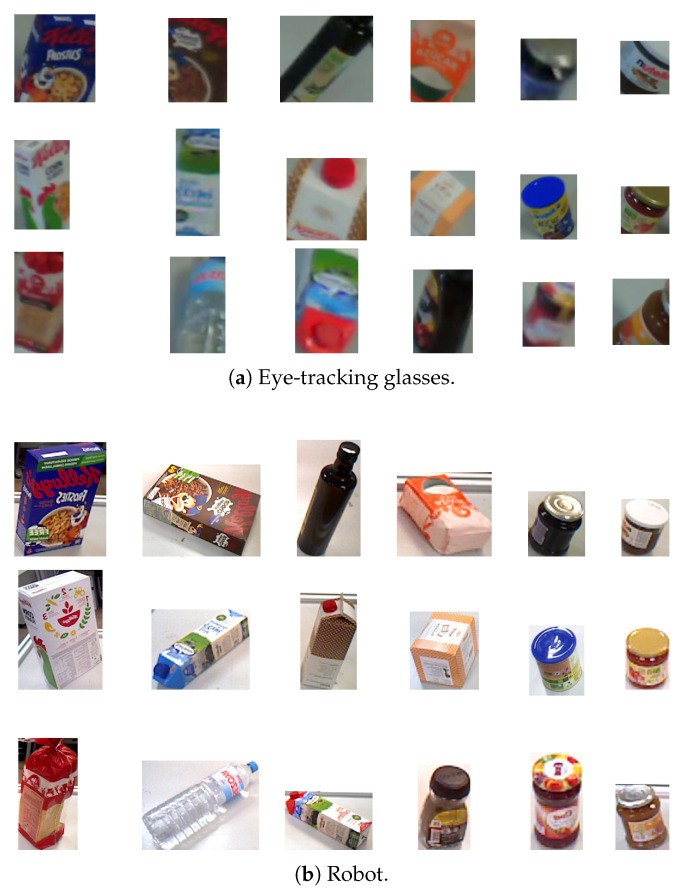
Dataset sample images. (**a**) Eye-tracking glasses images with motion artifacts. (**b**) Robot-taken images from a fixed position.

**Table 1 biomimetics-09-00100-t001:** Mean and standard deviation of superquadric parameters and average shape estimation error per object category.

Category	$\lambda_{1}$ [m]	$\lambda_{2}$ [m]	$\lambda_{3}$ [m]	$\lambda_{4}$	$\lambda_{5}$	Avg. Error [m]
cereal1	0.157±0.010	0.103±0.016	0.037±0.007	0.232±0.089	0.206±0.066	0.0063
cereal2	0.155±0.011	0.104±0.013	0.034±0.006	0.218±0.073	0.200±0.047	0.0062
milk1	0.128±0.006	0.040±0.005	0.033±0.005	0.315±0.098	0.277±0.093	0.0058
milk2	0.130±0.008	0.040±0.006	0.034±0.005	0.296±0.097	0.269±0.092	0.0051
jam1	0.057±0.005	0.035±0.002	0.034±0.005	0.448±0.079	0.363±0.184	0.0091
jam2	0.058±0.005	0.035±0.002	0.035±0.004	0.468±0.093	0.359±0.163	0.0087
jam3	0.069±0.005	0.032±0.002	0.030±0.003	0.499±0.072	0.344±0.047	0.0084
sugar1	0.059±0.009	0.056±0.011	0.048±0.009	0.232±0.075	0.392±0.127	0.0055
sugar2	0.117±0.007	0.041±0.006	0.036±0.005	0.231±0.073	0.228±0.090	0.0056
nutella	0.045±0.004	0.037±0.003	0.037±0.004	0.457±0.097	0.408±0.208	0.0071
olive-oil	0.144±0.011	0.023±0.002	0.018±0.003	0.405±0.086	0.349±0.057	0.0151
tomato-sauce	0.045±0.004	0.040±0.004	0.038±0.004	0.442±0.080	0.410±0.212	0.0108

**Table 2 biomimetics-09-00100-t002:** Common hyperparameters of the Siamese networks.

Hyperparameter	Value
Learning Rate	0.0001
Optimizer	Adam
Loss Function	Batch Hard Triplet Loss
Effective Batch Size	72
Regularization	Batch Normalization, Early Stopping

**Table 3 biomimetics-09-00100-t003:** Evaluation metrics of the trained models at best thresholds.

Feature Extractor Model	Threshold	Average Loss	Accuracy	Precision	Recall	F1 Score
ResNet-50	1.5970	0.0134	0.9712	0.8304	0.9646	0.89249
ResNet-18	1.3859	0.1029	0.69015	0.645	0.8452	0.732
VGG16	1.8232	0.2107	0.61075	0.5077	0.6895	0.59885
VGG19	1.9232	0.2008	0.63005	0.6067	0.7395	0.66655
MobileNet-V2	1.4818	0.0687	0.7247	0.6648	0.9061	0.7669

**Table 4 biomimetics-09-00100-t004:** Evaluation metrics of the Siamese network when using the mask based on superquadrics.

Superquadric-Based Mask	Threshold	Average Loss	Accuracy	Precision	Recall	F1 Score
NO	1.5970	0.0238	0.750	0.6843	0.928	0.7878
YES	1.5970	0.0194	0.8515	0.8122	0.9145	0.8603

## Data Availability

Data are contained within the article.

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
