# Peer review of "Integrating Egocentric and Robotic Vision for Object Identification Using Siamese Networks and Superquadric Estimations in Partial Occlusion Scenarios"

_biomimetics, 2024, doi:10.3390/biomimetics9020100_

Round 1

Reviewer 1 Report

Comments and Suggestions for Authors

The paper proposed an improved object identification method using Siamese networks and superquadric estimations based on both egocentric and robotic vision.  Interesting results are presented for the challenging partial occlusion cases.  The paper needs some revisions as follows:

1. Please include a table containing the hyperparameter values used in the Siamese networks.

2. Please add the comparison results. For instance, the authors may consider the performances of similar/existing method(s) for their comparison.

Reviewer 2 Report

Comments and Suggestions for Authors

Summary and understanding:

This paper tackles a combination of several Human-Robot Interaction related problems:

• class-agnostic object identification

• user-guided object identification

• intension estimation

• gaze event identification

The problem is basically identification (segmentation without classification) of objects that are the target of a user’s gaze, thus first the goal is intent estimation, then category-agnostic identification. The paper tackles an environment where the peripheral inputs are:

• RGB Camera

• Depth Camera

• Eye Tracker

When it comes to gaze estimation, previous methods often rely on prior knowledge of objects and their fixed positions or the use of markers which restricts their use in realistic environments (with unknown objects or obfuscated/obscure variations).

To solve the problem the authors trained Siamese Networks for identification of gazed objects, the dataset crops the segments of each category agnostic object based on its superquadratic, where superquadratics are a compact representation of simple objects with a set of parameters, thus achieving the category agnostic behaviour that is desired.

The workflow of the developed framework is the following:

• a depth image is acquired with the robot camera, then combined with the camera intrinsic

parameters and the robot joint configuration with forward kinematics, this depth image is

converted into a 3D point cloud, and the joints are taken out from the cloud.

• the point cloud is clustered.

• by projecting the points of each cluster onto the horizontal plane and computing the convex

hull of them, a reconstruction of occlusions is achieved.

• Each cluster is run through an inside-outside function in order to calculate its

superquadratic.

• Using a direct formulation on the surface of a superquadratic a mask is obtained for each

superquadratic.

• The RGB image is cropped based on the superquadratic mask for each object:

â—¦ a positive crop where the object is taken out.

â—¦ a negative crop where the other objects are taken out.

â—¦ an anchor image where the object is highlighted by its mask.

• a Triplet neural network is constructed, which extends the concept of Siamese Networks by learning embeddings based on the Three-Crops of each object, and the Triplet Loss Function is used to train the network, the network produces 3 outputs: positive, negative, and anchor.

• Through the fusion of the three outputs, object segmentation is achieved based on shape and pose rather than based on category.

In the results section the authors discuss their work by testing the average error for different types of objects, and under different circumstances (varying occlusions), they also test their work with superquadratics against variants that do not rely on superquadratics and they conclude that the superquadratics method achieves better numerical results, however no direct comparison with previous works is done.

Comments:

Great work, however it would be useful to provide an overarching diagram (with a small section explaining the workflow/dataflow) for how all the components of the discussed solution framework actually work in tandem together.

Reviewer 3 Report

Comments and Suggestions for Authors

This study aims to develop an optimization model that enables a robot to identify a selected object based on the user's gaze, facilitated by eye-tracking glasses. However, the results are limited and lack in-depth explanations and validation. Also, the novelty and significance of the manuscript are not clear. The authors should clearly explain the new contributions in this work to fill the research gaps. The manuscript has many mathematical equations that are out of the reader's interest because it is difficult to follow them. It needs in-depth discussions and explanations of equations and their benefits. In addition, the technical writing has many grammatical errors, making it hard to read. 

In addition to the following:

- The abstract is vague and broad. Enhance the abstract to emphasize this work's objectives by adding a paragraph about the findings and results.

- Enhance the conclusion to focus only on the objectives, methodology, and quantitative results. It should present the limitations and future directions of this study.

-Enhance the introduction to present sufficient knowledge and recent studies on the current problem.

- Add more literature review to focus on the current objectives of the proposal and identify the research Gaps. Add recent studies 2023, 2022, etc.

 - Most of the Figures missed clear explanation and analysis. For example, in Fig. 7, What does dimention2, 3,4,5 mean?

-The research paper should be written in the third person's perspective; words such as "we" must be avoided.

Also, how the model computes the proposed system efficiency compared to other studies needs to be clarified.

- Enhance the figure resolutions and captions (too long).

- Too long sentences make the meaning unclear and hard to read. Consider breaking it into multiple sentences, such as the following examples: L6-L8; L26-L28; L37-L40; L44-L47; 

The English language, redaction, and punctuation must be improved in general. The manuscript should undergo editing before being submitted to a journal again. The following are some examples:

L1: robot identifying a . …… should be …. robot to identify a

L3: the objects categories or their categories …… should be …. the objects' categories or categories

L3: without the use of external markers…… should be …. without external markers

L5: matches the user's gaze, with …… should be …. that matches the user's gaze with

L6: estimating objects shapes …… should be …. estimating objects' shapes 

L8: to the identification of new …… should be …. to identification new

Comments on the Quality of English Language

- Too long sentences make the meaning unclear and hard to read. Consider breaking it into multiple sentences, such as the following examples: L6-L8; L26-L28; L37-L40; L44-L47; 

The English language, redaction, and punctuation must be improved in general. The manuscript should undergo editing before being submitted to a journal again. The following are some examples:

L1: robot identifying a . …… should be …. robot to identify a

L3: the objects categories or their categories …… should be …. the objects' categories or categories

L3: without the use of external markers…… should be …. without external markers

L5: matches the user's gaze, with …… should be …. that matches the user's gaze with

L6: estimating objects shapes …… should be …. estimating objects' shapes 

L8: to the identification of new …… should be …. to identification new

Reviewer 4 Report

Comments and Suggestions for Authors

Overall, your introduction is well-written and effectively sets the stage for the Human-Robot Interaction (HRI) approach you are presenting. 

It provides a clear context for the importance of robots understanding and responding to human gaze in order to enhance interactions.How ever here some suggestion and recommendations for you

1. Emphasize why gaze is considered an intuitive and non-verbal method of communication. What unique advantages does it offer in the context of Human-Robot Interaction?

2. The closing sentence of the introduction could be strengthened. Perhaps you could briefly mention the overarching goal of the proposed strategy or highlight the innovation or the novelty it brings to the field of HRI

3. Figure 1. illustration of the interaction scenario. The user gazes at the red glass on a table, and the robot’s task is to identify and grasp it. 

   However, the robot faces a challenge due to its lack of prior knowledge about object categories and positions. And also lack of information technology and specification of the glasses and robot comera 

4. How come the systems work properly? and how to acquire and to process the data? 

Methodology and experimental procedures are equivocal. Please provides a concise and brief overview, detailing your approach and experimental findings for reproducibility

How does this method compare to existing gaze-based object identification approaches? Acknowledging and discussing related work would strengthen the paper's contribution.

Are there any limitations to the method? For example, how does it handle situations with multiple objects in the robot's view?

Reviewer 5 Report

Comments and Suggestions for Authors

This paper presents a novel method to enable a robot identifying a selected object based on the user’s gaze facilitated by eye-tracking glasses. Please resolve the following issues before accepting again.

1.     The control experiments in the paper are too rudimentary, and more models should be added, such as ResNet-18 and VGG16.

2. The real-time performance of the method proposed by the author is questionable. The author claims that the average processing time for a object is 44 ms, which means only about 22 frames per second. I think this is a huge challenge for real-time processing.

Round 2

Reviewer 1 Report

Comments and Suggestions for Authors

The paper has been greatly improved and can be accepted for publication. 

Author Response

Thank you for your supportive feedback.

Reviewer 3 Report

Comments and Suggestions for Authors

The author has responded to most of my comments. However, I suggested that the author should make a small comparison with any of the studies mentioned in the literature survey section. This comparison will help to demonstrate the power and innovation of the proposed system.

Avoid using many references together, such as L30: [3-5], etc. You should classify the studies and write a paragraph about each study or category.

Figure 8 needs to be modified to give the meaning of the used architecture or remove it.

What is the meaning of "d" in Figure 9? Please clarify it in the text.

Comments on the Quality of English Language

The English language, redaction, and punctuation are fair, but there is room for improvement before resubmitting to the journal.

see the two examples.

L18: Now they must   ..... should be ....  Now, they must 

L24: without the need for speech    ..... should be ....  without needing for 

Reviewer 4 Report

Comments and Suggestions for Authors

We sincerely appreciate the time and effort you dedicated to revising your manuscript.

You have revised your paper properly. 

Please, consider adding a sentence or phrase that explicitly highlights the potential benefits or implications of this method for end-users or the broader field of Human-Robot Interaction. Also, adding a sentence about potential future directions or applications stemming from this research 

Reviewer 5 Report

Comments and Suggestions for Authors

Accept

Author Response

Thank you for your supportive feedback.